# Evaluation of Different Thawing Protocols on Iberian Boar Sperm Preserved for 10 Years at Different Liquid Nitrogen Levels

**DOI:** 10.3390/ani14060914

**Published:** 2024-03-15

**Authors:** Manuel Álvarez-Rodríguez, Cristina Tomás-Almenar, Helena Nieto-Cristóbal, Eduardo de Mercado

**Affiliations:** 1Department of Animal Reproduction, Spanish National Institute for Agricultural and Food Research and Technology (INIA-CSIC), 28040 Madrid, Spain; manuel.alvarez@inia.csic.es (M.Á.-R.); helena.nieto@inia.csic.es (H.N.-C.); 2Agro-Technological Institute of Castilla y León (ITACyL), Ctra. de Burgos Km. 119, Finca Zamadueñas, 47071 Valladolid, Spain; tomalmcr@itacyl.es

**Keywords:** long-term storage, Iberian pig semen, cryopreservation, sperm quality

## Abstract

**Simple Summary:**

It remains unclear whether the prolonged preservation of cells, such as sperm, in liquid nitrogen (LN_2_) leads to their deterioration over time. In this study, we investigated whether the conservation of Iberian pig semen for ten years results in a decline in quality and whether the evaporation of LN_2_ from the tanks between refills could be a contributing factor. Furthermore, the only way to improve already-frozen samples is to implement specific thawing protocols. Thus, our findings suggest that only samples consistently submerged in LN_2_ benefit the most from the application of thawing protocols, such as rapid thawing at 70 °C for 8 s. While partial conservation in LN_2_ and LN_2_ vapors does not result in a loss of quality over time, it does cause damage that thawing protocols cannot mitigate. This finding suggests that, independently of the lack of the effect of the level of storage in the tank, careful handling and optimization of thawing protocols are essential for the quality preservation of the cryopreserved sperm samples.

**Abstract:**

The conservation of genetic resources in pig breeds, notably the Iberian pig, is crucial for genetic improvement and sustainable production. Prolonged storage in liquid nitrogen (LN_2_) is recognized for preserving genetic diversity, but potential adverse effects on seminal quality remain debated. This study aims to assess the impact of ten years of storage at different LN_2_ levels and to optimize thawing protocols for Iberian pig sperm. Sperm samples from 53 boars were cryopreserved and stored at varying LN_2_ levels and, a decade later, the samples were thawed at 37 °C for 20 s or at 70 °C for 8 s. Sperm motility, membrane integrity, acrosome status, and DNA fragmentation were evaluated in year 0 and year 10. Overall, no significant differences were observed in post-thaw sperm quality between storage levels in year 0 or year 10. But thawing at 70 °C 8 s showed significant improvements, particularly in samples that were always stored in LN_2_, in all analyzed parameters except fragmentation, which was not affected by cryostorage. This study suggests that the long-term preservation of Iberian pig sperm does not affect quality over time, regardless of whether the samples were fully submerged in LN_2_. Furthermore, it is determined that thawing at 70 °C for 8 s maximizes post-thaw sperm quality, especially in those samples stored constantly submerged in LN_2_.

## 1. Introduction

The conservation of animal genetic resources plays a pivotal role in the preservation and enhancement of animal breeds, as well as in ensuring food security and the sustainability of pig production. The preservation of genetic material contributes to genetic improvement, enabling the selection of animals with superior characteristics such as disease resistance, improved feed efficiency, and enhanced productive performance.

The Iberian pig stands out as one of Spain’s most prominent pig breeds, representing a significant position in the country’s culture, economy, and gastronomy. Modern swine production systems, facilitated by artificial insemination (AI), have successfully reduced the number of male animals relative to females [1]. This, coupled with the prevalent practice of crossbreeding a pure Iberian sow with a male Duroc breed, has led to a decline in genetic diversity, particularly among males within the Iberian breed.

The preservation of sperm cells is recognized as an indispensable strategy to uphold genetic diversity and avert the loss of desirable traits in pig breeds [2]. Consequently, it is a valuable tool in conserving the variability inherent to the Iberian breed. In this sense, long-term storage of sperm cells in liquid nitrogen (LN_2_) has demonstrated its effectiveness in maintaining the availability and viability of genetic material across time [3,4,5,6,7]. Nevertheless, recent published studies have highlighted the potential adverse effects of extended storage in LN_2_ on seminal quality [8,9,10,11]. Few research studies have been conducted to assess the impact of prolonged semen storage in LN_2_, particularly within the porcine species [5,9,11]. In these studies, semen samples remained submerged in LN_2_ but stored in banks with regular handling, which might entail abrupt temperature fluctuations due to the handling of straws in and out of LN_2_ [12].

Storage tanks for straws typically come in varying heights, with superior compartments being more vulnerable to non-submersion if LN_2_ levels drop. In larger storage tanks equipped with temperature sensors, the internal temperature remains stable at −170 °C if the LN_2_ level remains above 50%. Previous observations indicate that this temperature is the most suitable for preserving most tissues [13,14]. However, even in scenarios where the internal tank temperature is maintained, samples from the higher levels transition from complete submersion in LN_2_ to a gaseous environment in relation to a decrease in the LN_2_ level.

In established storage systems, once the samples are submitted to the cryopreservation procedure, the only way to maximize semen quality is by modulating the thawing procedure. Nevertheless, the exploration of thawing procedures is scarce in the literature. The routine procedure involves plunging a straw in a circulating water bath at 37 °C for 20 s and dilution 1:2 in Beltsville Thawing Solution (BTS). A previous study tested modifications in the composition of thawing extenders and in thawing rates that lead to enhancements in post-thawing quality, improving the results from conventional thawing protocols [15].

Consequently, and taking into account two relevant aspects of sperm cryopreservation, the storage levels and the thawing procedures, the aim of this study was (i) to analyze the Iberian pig sperm quality preserved for ten years at different LN_2_ levels and (ii) to evaluate if modifications in the thawing protocol can mitigate the potential deleterious effect of prolonged preservation. To address our hypothesis, our experimental approach included the analysis of sperm motility and kinetics, membrane integrity, acrosomal status, and DNA fragmentation, all of which are relevant analysis tools for sperm quality assessment.

## 2. Materials and Methods

### 2.1. Ethics Statement

The ejaculates used for this study on their collection day complied with European (2010-63-EU; ES13RS04P, July 2012) and Spanish (ES300130640127, August 2006) regulations and rules regarding the commercialization of seminal AI doses, animal welfare, and health.

### 2.2. Reagents and Media

All chemicals and reagents were purchased from Sigma Chemical Co., (St. Louis, MO, USA). Orvus ES Paste is marketed as Equex STM, Nova Chemical Sales Inc., Scituate, MA, USA.

The extension and thawing extender was Beltsville Thawing Solution (BTS: 205 mM glucose, 20.39 mM NaCl, 5.4 mM KCl, 15.01 mM NaHCO_3_, 3.35 mM EDTA, pH 7.2, and 290–300 mOsmol/kg [16], containing kanamycin sulfate (50 mg/mL). The freezing extender was lactose–egg yolk (LEY) (80% [*v*/*v*] 310 mM β-lactose, 20% [*v*/*v*] egg yolk, 100 µg/mL kanamycin sulfate; pH 6.2, and 330 ± 5 mOsm/kg).

### 2.3. Collection and Freezing–Thawing Procedure of Sperm Samples

For this study, ejaculates were collected using the gloved-hand method from 53 healthy and fertile different boars of the Iberian breed that were frozen. All the animals used for this study were in the age range of 1.5 to 2 years and were housed in several Spanish artificial insemination centers with environmental control and fed with isoprotein and isoenergetic diets according to their breed. All ejaculates underwent the same processing before freezing in terms of collection and transportation. Briefly, after collection, the ejaculates were extended (1:1, *v*/*v*) in BTS and cooled to 17 °C. Once this temperature was reached, the ejaculates were transported inside insulated thermal boxes (17 °C) to the laboratory for further processing. Only ejaculates with ≥200 × 10^6^ sperm/mL, ≥85% sperm with normal morphology, and ≥75% and ≥80% of motile and viable spermatozoa, respectively, were selected for cryopreservation. Only one ejaculate per boar was cryopreserved using the straw-freezing procedure originally described by [17] and subsequently modified by [18,19]. In brief, ejaculates were obtained using the gloved hand method and diluted with BTS (*v*/*v*). Samples were centrifuged at 2400× *g* for 3 min at 15 °C, and the pellet was extended in LEY to reach a concentration of 1.5 × 10^9^ cells/mL. After further cooling to 5 °C within 120 min, the diluted spermatozoa were resuspended in LEY-glycerol-Orvus ES Paste (LEYGO) extender (92.5% LEY, 1.5% Equex STM [Nova Chemical Sales Inc., Scituate, MA, USA] and 6% glycerol (*v*/*v*); pH 6.2, and 1650 ± 15 mOsm/kg) to yield a final concentration of 1 × 10^9^ cells/mL. The resuspended and cooled spermatozoa were packed into 0.5 mL PVC-French straws, which were frozen using a controlled-rate freezer as follows: from 5 to −5 °C at a rate of 6 °C/min, from −5 to −80 °C at 40 °C/min, held for 30 s at −80 °C, then cooled at 70 °C/min to −150 °C and plunged into LN_2_.

The thawing process was carried out as specified in the experimental design section. All samples were incubated in a water bath at 37 °C for up to 30 min, the time at which post-thaw sperm quality was assessed.

### 2.4. Experimental Design

After cryopreservation, two straws from each of the 53 boars (9 boars at Level 1, 17 boars at Level 2, and 27 boars at Level 3) were thawed one week later at 37 °C for 20 s (Year 0). In year 0, motility was analyzed immediately after dilution of the samples. The remainder was stored in an Arpege 170 straw storage tank (Air Liquide) for ten years without any intervening extraction. After this elapsed duration (Year 10), the same number of animals and straws were subjected to analysis upon thawing in two distinct manners. The first method mirrored the initial procedure conducted a decade ago (37 °C for 20 s), while the second involved thawing the samples at 70 °C for 8 s, but in both there was a 10 min incubation at 37 °C after dilution with BTS for the CASA analysis. 

Over those ten years, the stored sperm of each male remained in a fixed position without being repositioned or removed from the tank. These samples were distributed across various levels (1 to 3 in Figure 1) within the racks of the storage tank, resulting in three distinct levels from the tank’s entrance. LN_2_ within this tank was consistently replenished whenever its level approached slightly above 50%, approximately every two months, ensuring that the interior temperature remained constant, automatically tracked by a built-in sensor. The LN_2_ replenishment, driven by the tank’s inherent evaporation, led to the following three exposure conditions for the samples: the samples positioned at the upper level (*n* = 9 boars) were submerged in LN_2_ for one month and exposed to LN_2_ vapors for one month (approximately 60 months in LN_2_ and 60 months in vapors in the ten years of the study); samples at level 2 (*n* = 17 boars) were partially submerged for one month during each refill (approximately 30 months partially submerged in LN_2_ in the 10 years of the study); while samples at level 3 (*n* = 27 boars) remained consistently immersed in LN_2_ (Figure 1).

### 2.5. Sperm Evaluation

#### 2.5.1. Sperm Motility

A computer-assisted sperm analysis system (ISAS^®^, version 1.0; Proiser, Valencia, Spain) was employed to objectively assess sperm motility characteristics in this study. The system operated at a frame rate of 25 video frames per second (25 Hz). Semen samples were rediluted in BTS to achieve a concentration of 30 × 10^6^/mL and incubated for 10 min at 37 °C, but only in the samples assessed at 10 years. For each evaluation, 3 µL of the sperm sample was deposited into a pre-warmed (38 °C) Makler counting chamber. A minimum of 300 sperm (across four to five fields) were recorded for each sample. The following sperm motility parameters were evaluated: total motile sperm (TMS, %), progressive motility sperm (PMS, %), curvilinear velocity (VCL, µm/s), straight-line velocity (VSL, µm/s), average path velocity (VAP, µm/s), percentage of linearity (LIN, %), percentage of straightness (STR, %), wobble coefficient (WOB, %), mean amplitude of lateral head displacement (ALH, mm), and mean beat cross frequency (BCF, Hz).

#### 2.5.2. Plasma Membrane Integrity Assessment

Sperm plasma membrane integrity was evaluated using phase contrast microscopy (Nikon Eclipse E400, Tokyo, Japan) coupled with a fluorescence system (Nikon C-SHG1) equipped with a mercury lamp (100 W) and a Nikon filter (510/590 nm), following a previous technique [15]. An aliquot was diluted with BTS to 30 × 10^6^ sperm/mL. Then, each sample was stained with 5 μL of propidium iodide (PI, 0.5 mg/mL) and incubated at 37 °C in darkness for 10 min. The samples were examined simultaneously under phase contrast microscopy and fluorescence. Non-viable sperm exhibited a red coloration (PI+), while viable sperm remained unstained and were discernible through phase contrast. A minimum of 300 cells per sample in random fields were examined, and the data are presented in the results as a percentage of sperm with intact plasma membrane (SIPM, %) (non-red stained sperm by PI).

#### 2.5.3. Acrosome Status Assessment

The acrosome status was evaluated by phase contrast microscopy at 1000×. Briefly, samples were fixed in 2% glutaraldehyde, and a minimum of 200 acrosomes were examined per sample. The damage to the acrosome cap was classified by the scoring system previously reported by [15], and the data are presented in the results as a percentage of spermatozoa with a normal acrosomal ridge (NAR, %)

#### 2.5.4. DNA Fragmentation

To assess the DNA fragmentation index (DFI) in boar sperm cells, we employed the Sperm Sus-Halomax kit (Halotech, Madrid, Spain). This kit uses the sperm chromatin dispersion technique, which is grounded in the distinct response of sperm chromatin to a protein depletion treatment [20]. 

Briefly, diluted sperm samples were combined with agarose in a vial. Pre-treated slides provided with the kit were maintained at 4 °C. Subsequently, 2 µL of the cell suspension were evenly applied to the treated side of the slide and covered with a glass coverslip for 5 min at 4 °C. Following this, the coverslip was gently removed, and the samples were treated with the lysing solution in the kit. The slides were then washed for 5 min and dehydrated in sequential baths of 70%, 90%, and 100% ethanol. Finally, samples were stained with fluoRed^®^ fluorescence microscopy staining provided by the kit. The spermatozoa were evaluated at 40× under a fluorescence microscope. A minimum of 300 spermatozoa were analyzed. Spermatozoa with fragmented DNA showed nucleoids with a large and spotty halo of chromatin dispersion (FRAG, %). In contrast, those without fragmented DNA showed nucleoids with a small and compact halo of chromatin dispersion.

### 2.6. Statistical Analysis

The statistical analysis was conducted by analysis of variance (ANOVA) after testing the normal distribution of the data using the Shapiro–Wilk test and Levene’s test. When the ANOVA showed differences, Tukey’s test was used for multiple comparisons between means with a statical significance level of 0.05 (SPSS 29 (IBM, Armonk, NY, USA)). The effect of the position of the straws in the tank (level 1, level 2, and level 3) was analyzed at year 0 and at year 10 for both thawing conditions. The effects of time and thawing conditions were evaluated at the same level. Data are shown as mean ± standard error of the mean (SEM).

## 3. Results

### 3.1. Comparisons within the Same Year and Thawing Process

The results from the year 0 analysis (Figure 2, Table 1) revealed no statistically significant differences (*p* < 0.05) in any of the post-thawing quality parameters assayed across various storage levels. In addition, the post-thaw sperm quality results for year 10, independently of the post-thawing protocol used (37 °C or 70 °C), did not reveal any significant differences (*p* < 0.05) among the storage levels (Figure 2, Table 1).

### 3.2. Comparisons within the Same Levels across Different Years and Thawing Methods

#### 3.2.1. Level 1

The sperm quality parameters TMS, PMS, VCL, VSL, VAP, LIN, STR, and WOB were significantly different (Figure 3, Table 2, *p* < 0.05) between Year 0 and Year 10, regardless of the thawing temperature employed, for level 1 of storage. However, no significant differences were found in SIPM, NAR, Frag (Figure 3), ALH, and BCF (Table 2).

#### 3.2.2. Level 2

The results observed in level 2 over different time points showed a similar pattern to level 1, but the most pronounced difference was in the experimental group thawed at 70 °C, which displayed a significantly higher SIPM than the other two groups (Figure 4). Thus, a significant difference was found between the Year 10 at 70 °C group and the Year 0 group in terms of total and progressive motility.

Furthermore, the parameters related to sperm kinetics mirrored those in level 1, with the Year 0 group consistently demonstrating the highest values (VCL, VSL, VAP, LIN, STR, and WOB; Table 3).

#### 3.2.3. Level 3

Data collected at level 3 exhibited that in this context, the highest SIPM and NAR values were found in the Year 10 group, subjected to thawing at 70 °C, vs. the Year 0 group (Figure 5).

Sperm motility parameters presented significant variations among the three groups, with the Year 10 group at 70 °C registering the highest values for both TMS and PMS (Figure 5). Interestingly, the outcomes related to sperm kinetics remained consistent with those at the other levels. However, it’s worth noting that the Year 10 at 70 °C group exhibited VCL equivalent to the other two groups and ALH greater than that of the Year 0 group (Table 4).

## 4. Discussion

The available literature on evaluating the impact of prolonged semen preservation on the seminal quality in swine is very scarce, and, to the best of our knowledge, this is the first study in the Iberian pig breed. In this sense, there is a significant controversy regarding whether long-term preservation has a detrimental effect on seminal samples. Discrepancies exist in the literature, with some authors addressing a detrimental impact [8,9,10,11], while others did not find such an effect [3,4,5,6,7]. In any case, the underlying causes related to this deleterious effect remain poorly elucidated. Our study aimed to evaluate whether prolonged storage may be influenced by the specific arrangement of the samples within the storage tanks, as those at higher levels are more exposed to temperature fluctuations, LN_2_ losses, and increased handling-related potential problems. Our results showed no differences in any parameter analyzed when it comes to the same thawing protocol used at different levels both in year 0 and after 10 years of cryostorage, so their particular position in the tank does not influence sperm quality after thawing. These results may suggest that extended preservation had no detrimental effect, despite varying conditions across the levels. Furthermore, we tested that modifications in thawing protocols could assist in mitigating or reducing any potential harm that might arise from this storage. Our results confirmed that the beneficial effects are directly linked to the storage level, with the lower levels showing a clear beneficial effect of modifying the thawing temperature.

Potential factors contributing to damage during prolonged preservation may be associated with sample handling and maintenance practices. Despite submerging the samples in LN_2_, the regular manipulation of the tank, involving the input and output of straws, can lead to abrupt temperature fluctuations. Even if the temperature drop did not lead to the thawing of the samples, repeated exposure to higher temperatures can result in a decline in sperm vitality [3]. Some studies did not include specific information about storage conditions [3,4,5,6,7], while others provided detailed information about storage conditions, ensuring that the samples were always fully immersed in LN_2_ [8,9,10,11]. Interestingly, these last studies have reported adverse effects of prolonged preservation. However, our results from 10 years of storage showed no differences in a crucial factor, sperm membrane intactness, relative to year 0.

Among the parameters widely described as most affected by extended preservation is motility, with significant reductions observed after two years of storage [8,9,10,11]. In our study, motility was not affected by storage, as depicted by results comparing year 0 with year 10. In fact, modifying the thawing protocol temperature lead to an increase in these parameters at all tank levels. Motility is a crucial parameter of seminal quality closely linked to fertility [21]. However, its quantification is subject to various external factors that can influence it [22,23]. Previous results from our group have demonstrated that modifications in thawing media or the thawing protocol can significantly enhance motility compared to standard thawing protocols [15,24]. In agreement with those studies, the current study showed a significant increase in both total and progressive motility in the experimental group using 70 °C, relative to year 0. In addition, thawing at 70 °C had a marked beneficial effect at level 3, where total motility closely resembling a similar pattern of viability was recorded. This motility recovery has been observed in previous studies through modifications in thawing rates [15], most likely linked to avoiding the growth of ice crystals during recrystallization, which can cause severe damage to sperm [25]. Although we did observe changes in movement quality parameters between year 0 and year 10, with lower velocities, as noted in other swine studies [10], we did not experience a decrease in BCF, a parameter strongly associated with litter size and farrowing rate prediction [26,27].

Our study incorporates a fundamental modification in the thawing process at the ten-year point, which involves post-dilution incubation of the samples. It has been established that dilutions below 20 × 10^6^ sperm/mL can decrease motility due to the dilution effect [28,29]. However, for accurate visualization in the Computerized Semen Analysis (CASA) system, proper dilution is essential to prevent sperm collisions and disruptions in their movements. Our dilution is close to the threshold that may induce this decline in motility and might be related to the lower motility observed in year 0 compared to year 10. On the other hand, the 37 °C incubation in the new dilution medium for 10 min managed to prevent or reverse the motility loss. Since there was no change in temperature, this effect could be related to acclimatization, akin to what is observed when performing a 1:2 dilution before creating semen doses [30]. The mechanisms underlying this acclimatization are not fully understood but may be associated with the reorganization of membrane lipids and proteins [31]. Moreover, in all studies where a motility decline over time has been observed, even in their year 0, there is a difference of at least 30–40% between live and motile sperm. Our study, in line with other results [15,24], suggests that optimizing the thawing protocol can lead to increased motility, resulting in a total motile percentage similar to viable sperm. Thus, a potential improvement in motility losses observed by other authors could be obtained through the optimization of incubation, thawing rates, or modifications in the thawing medium.

Acrosomal status and membrane integrity are another fundamental parameter of semen quality assessment tracked in our study. In both cases, significant improvements were only observed with the change in the thawing rate, but only in samples stored entirely in LN_2_ and only partially in those stored at a medium LN_2_ level (level 2). These high thawing temperatures allow for swift passage through the recrystallization phase (from 50 °C to 0 °C) [32,33,34], a critical zone where ice crystal formation can cause sperm to transition directly from the glassy state to the liquid state [35], thus preventing cellular damage. There is also a rapid transition through the ‘warm shock’ temperatures (0 °C to 15 °C), which is the reverse process of ‘cold shock’ [36] during re-warming. This shock helps reduce damage to the membranes and acrosomes. Observing this improvement only in the lower levels could be attributed to the fact that the temperature inside the tank indeed experienced variations depending on the LN_2_ level, even though the sensors in the closed tank might indicate a constant temperature throughout. It has been determined that the decrease in LN_2_ levels in the tanks significantly raises the temperature in areas closer to the neck of the tank [12]. Furthermore, it has also been observed that, depending on the tank level, manipulating the straws when raising and lowering them can lead to drastic temperature changes in the straws, even reaching the recrystallization temperature of −130 °C [12,37]. In our study, the samples were not moved, meaning they remained at a consistent height. However, as seen in the studies above, samples from the upper levels may have experienced a gradual temperature increase as the LN_2_ level decreased, which would reverse on the day the tank was refilled. This could imply that there may have been processes of de-crystallization and recrystallization, particularly in level 1, less so in level 2, and none in level 3. These processes do not cause lethal damage to sperm, as evidenced by the absence of significant differences in acrosome state, viability, or fragmentation after ten years in level 1. However, they cause damage that cannot be reversed or avoided during thawing. This damage could potentially be exacerbated or mitigated based on the lipid composition of sperm membranes. Research has revealed differences in phospholipid composition between “good” and “bad” freezers [38]. Thus, it would be beneficial for future studies to conduct lipidomics analyses to ascertain the impact of phospholipids on the long-term preservation process.

A noteworthy aspect of our study is the evaluation of DNA fragmentation. Some studies have shown that prolonged cryostorage increased sperm DNA fragmentation [8]. It has been determined that a possible cause of fragmentation damage is due to the oxidative stress that sperm undergo during freezing [39,40], and the increase or change in temperature that could occur during the storage and handling of the frozen samples may be a contributor to this phenomenon. However, our results have shown that DNA remained stable regardless of the sample’s storage level and thawing protocol, indicating no damage during storage. Furthermore, the observed fragmentation values, even from year 0, were very low, averaging less than 2%, confirming previous observations of relatively low levels of nuclear DNA fragmentation in thawed boar sperm [41].

## 5. Conclusions

Our results about the long-term preservation of Iberian pig sperm showed no differences, regardless of whether the samples were fully submerged in LN_2_. Furthermore, temperature modification in the thawing protocol, using 70 °C for 8 s, is recommended to maximize post-thaw sperm quality. This is the most relevant improvement for the samples stored at lower levels in the tank, which means that they are constantly submerged in LN_2_. While modifying the thawing protocol yields promising results, its implementation in production systems poses challenges due to the difficulty of handling samples at high temperatures within a short timeframe. Nevertheless, it is a valuable tool for the future, as the only means to enhance samples already cryopreserved in germplasm banks is by applying suitable thawing protocols. This underscores the need for future research efforts to optimize existing thawing systems, focusing on automation that streamlines and accelerates the thawing process. Even so, we should take special care of other factors that are more difficult to control. Excessive handling, abrupt temperature changes, or extended exposure to LN_2_ vapor alone may cause direct or sublethal damage that cannot be reversed during thawing.

## Figures and Tables

**Figure 1 animals-14-00914-f001:**
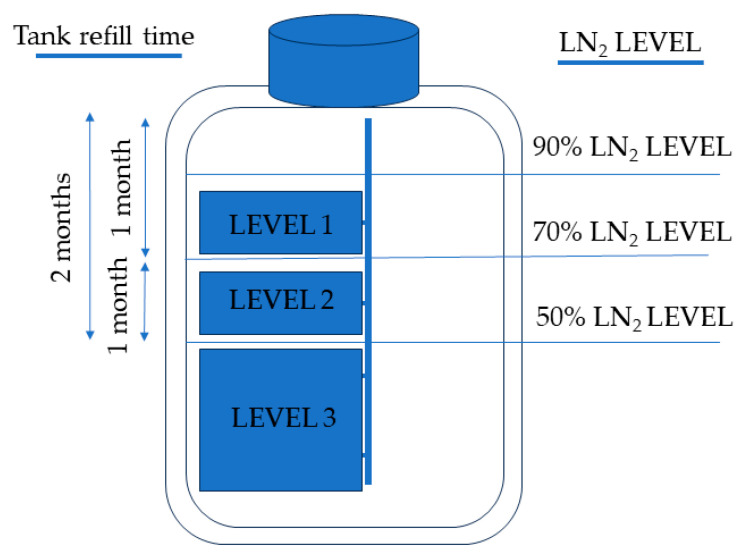
The diagram illustrates the relationship between the liquid nitrogen (LN_2_) levels in the storage tank and their corresponding storage levels. It also provides information regarding the frequency of tank refilling and the duration for which the straws were submerged in LN_2_. This graphical representation summarizes how the LN_2_ levels fluctuated in correlation with the storage levels and the specific intervals at which the tank required replenishment.

**Figure 2 animals-14-00914-f002:**
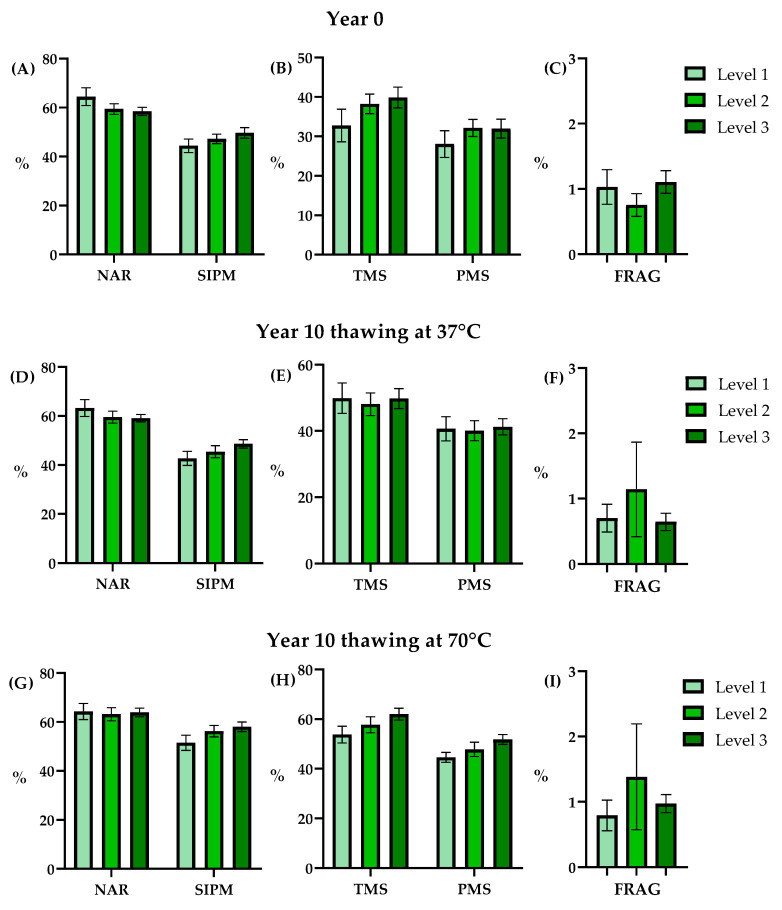
Semen quality parameters analyzed post-thaw between levels (Level 1: light green bar (*n* = 9); Level 2: green bar (*n* = 17); Level 3: dark green bar (*n* = 27)) in year 0. (**A**): normal acrosomal ridge (NAR, %) and sperm with intact plasma membrane (SIPM, %), (**B**): total motile sperm (TMS, %) and progressive motility sperm (PMS, %), (**C**): Spermatozoa with fragmented DNA (FRAG, %), in year 10 thawing at 37 °C, (**D**): normal acrosomal ridge (NAR, %) and sperm with intact plasma membrane (SIPM, %), (**E**): total motile sperm (TMS, %) and progressive motility sperm (PMS, %), (**F**): Spermatozoa with fragmented DNA (FRAG, %), and %, in year 10 thawing at 70 °C, (**G**): normal acrosomal ridge (NAR, %) and sperm with intact plasma membrane (SIPM, %), (**H**): total motile sperm (TMS, %) and progressive motility sperm (PMS, %), (**I**): Spermatozoa with fragmented DNA (FRAG, %). Data are depicted as means ± SEM.

**Figure 3 animals-14-00914-f003:**
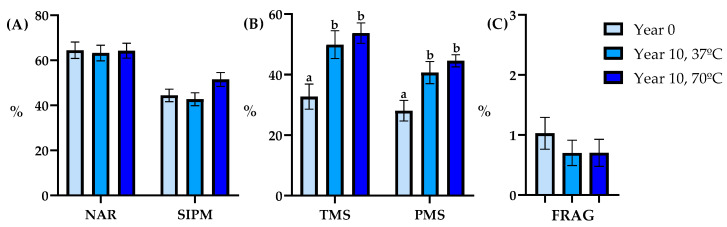
Semen quality parameters analyzed post-thaw between years and thawing methods (Year 0: light blue bar; Year 10, 37 °C: blue bar; Year 10, 70 °C: dark blue bar; *n* = 9) in Level 1. (**A**): normal acrosomal ridge (NAR, %) and sperm with intact plasma membrane (SIPM, %), (**B**): total motile sperm (TMS, %) and progressive motility sperm (PMS, %), (**C**): spermatozoa with fragmented DNA (FRAG, %). Data are depicted as Means ± SEM. Different lowercase letters indicate differences among groups (*p* < 0.05).

**Figure 4 animals-14-00914-f004:**
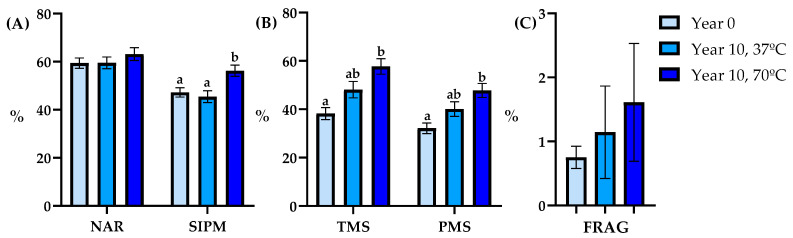
Semen quality parameters analyzed post-thaw between years and thawing methods (Year 0: light blue bar; Year 10, 37 °C: blue bar; Year 10, 70 °C: dark blue bar; *n* = 17) in Level 2. (**A**): normal acrosomal ridge (NAR, %) and sperm with intact plasma membrane (SIPM, %), (**B**): total motile sperm (TMS, %) and progressive motility sperm (PMS, %), (**C**): spermatozoa with fragmented DNA (FRAG, %). Data are depicted as Means ± SEM. Different lowercase letters indicate differences among groups (*p* < 0.05).

**Figure 5 animals-14-00914-f005:**
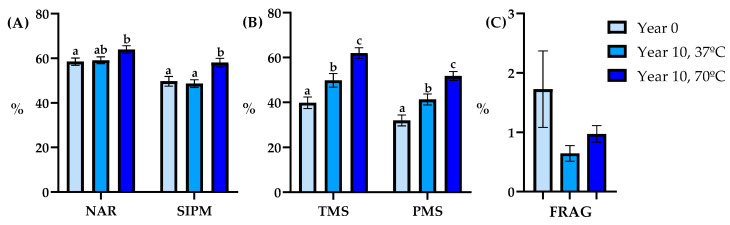
Semen quality parameters analyzed post-thaw between years and thawing methods (Year 0: light blue bar; Year 10, 37 °C: blue bar; Year 10, 70 °C: dark blue bar; *n* = 27) in Level 3. (**A**): normal acrosomal ridge (NAR, %) and sperm with intact plasma membrane (SIPM, %), (**B**): total motile sperm (TMS, %) and progressive motility sperm (PMS, %), (**C**): spermatozoa with fragmented DNA (FRAG, %). Data are depicted as Means ± SEM. Different lowercase letters indicate differences among groups (*p* < 0.05).

**Table 1 animals-14-00914-t001:** Kinematic parameters of motility analyzed post-thaw between levels (Level 1 (*n* = 9), Level 2 (*n* = 17), and Level 3 (*n* = 27)) in year 0 and year 10 thawing at 37 °C and in year 10 thawing at 70 °C.

Parameters	VCL	VSL	VAP	LIN	STR	WOB	ALH	BCF
	**Year 0**							
Level 1	112.7 ± 4.9	99.3 ± 3.4	99.3 ± 4.3	75.9 ± 1.8	86.0 ± 1.5	88.1 ± 0.8	3.0 ± 0.2	8.0 ± 0.2
Level 2	98.8 ± 4.4	86.0 ± 4.0	86.0 ± 3.9	71.9 ± 2.0	82.5 ± 2.0	87.0 ± 0.8	2.7 ± 0.1	7.6 ± 0.2
Level 3	99.1 ± 4	87.0 ± 2.9	87.0 ± 3.4	72.4 ± 1.8	82.2 ± 1.7	88.0 ± 0.7	2.7 ± 0.1	7.6 ± 0.1
*p*-value	0.18	0.05	0.14	0.52	0.5	0.6	0.3	0.3
	**Year 10 (37 °C)**							
Level 1	95.2 ± 1.5	54.8 ± 2.2	78.0 ± 1.3	57.8 ± 2.8	70.4 ± 3.0	81.9 ± 1.0	3.1 ± 0.1	7.9 ± 0.2
Level 2	84.2 ± 2.6	50.0 ± 1.5	69.0 ± 2.1	60.0 ± 2.0	73.0 ± 2.1	82.1 ± 1.0	2.8 ± 0.1	7.7 ± 0.2
Level 3	83.9 ± 2.8	50.9 ± 1.3	69.4 ± 2.4	61.6 ± 1.7	74.6 ± 1.9	82.6 ± 0.9	2.8 ± 0.1	8.0 ± 0.1
*p*-value	0.05	0.18	0.08	0.49	0.5	0.87	0.09	0.29
	**Year 10 (70 °C)**							
Level 1	93.9 ± 3.8	54.7 ± 2.0	74.9 ± 3.2	58.7 ± 2.3	73.7 ± 2.7	79.7 ± 1.0	3.2 ± 0.1	7.5 ± 0.2
Level 2	89.9 ± 2.7	51.6 ± 1.7	72.1 ± 2.3	57.6 ± 1.6	71.8 ± 1.7	80.1 ± 0.9	3.1 ± 0.1	7.6 ± 0.1
Level 3	90.3 ± 2.9	54.2 ± 1.4	73.3 ± 2.4	60.6 ± 1.4	74.7 ± 1.4	81.2 ± 0.6	3.0 ± 0.1	7.7 ± 0.1
*p*-value	0.73	0.41	0.81	0.36	0.48	0.43	0.45	0.77

Curvilinear velocity (VCL, µm/s), straight-line velocity (VSL, µm/s), average path velocity (VAP, µm/s), percentage of linearity (LIN, %), percentage of straightness (STR, %), wobble coefficient (WOB, %), mean amplitude of lateral head displacement (ALH, mm), and mean beat cross frequency (BCF, Hz). Data are depicted as means ± SEM.

**Table 2 animals-14-00914-t002:** Kinematic parameters of motility analyzed post-thaw between years and thawing methods (Year 0; Year 10, 37 °C; Year 10, 70 °C; *n* = 9) in Level 1.

	Year 0	Year 10, 37 °C	Year 10, 70 °C
VCL	111.6 ± 4.5 ^a^	95.2 ± 1.5 ^b^	93.9 ± 3.8 ^b^
VSL	82.2 ± 4.2 ^a^	54.8 ± 2.2 ^b^	54.7 ± 2.0 ^b^
VAP	96.2 ± 4.9 ^a^	78.0 ± 1.3 ^b^	74.9 ± 3.2 ^b^
LIN	73.8 ± 2.6 ^a^	57.8 ± 2.8 ^b^	58.7 ± 2.3 ^b^
STR	85.6 ± 1.4 ^a^	70.4 ± 3.0 ^b^	73.7 ± 2.7 ^b^
WOB	86.0 ± 2.2 ^a^	81.9 ± 1.0 ^ab^	79.7 ± 1.0 ^b^
ALH	3.1 ± 0.2	3.1 ± 0.1	3.2 ± 0.1
BCF	8.1 ± 0.2	7.9 ± 0.2	7.5 ± 0.2

Curvilinear velocity (VCL, µm/s), straight-line velocity (VSL, µm/s), average path velocity (VAP, µm/s), percentage of linearity (LIN, %), percentage of straightness (STR, %), wobble coefficient (WOB, %), mean amplitude of lateral head displacement (ALH, mm), and mean beat cross frequency (BCF, Hz). Data are depicted as means ± SEM. Different lowercase letters indicate differences among groups (*p* < 0.05).

**Table 3 animals-14-00914-t003:** Kinematic parameters of motility analyzed post-thaw between years and thawing methods (Year 0; Year 10, 37 °C; Year 10, 70 °C; *n* = 17) in Level 2.

	Year 0	Year 10, 37 °C	Year 10, 70 °C
VCL	98.8 ± 4.4 ^a^	84.2 ± 2.6 ^b^	89.9 ± 2.7 ^ab^
VSL	71.2 ± 4.0 ^a^	50.0 ± 1.6 ^b^	51.6 ± 1.7 ^b^
VAP	86.0 ± 3.9 ^a^	69.0 ± 2.1 ^b^	72.1 ± 2.3 ^b^
LIN	71.9 ± 2.0 ^a^	60.0 ± 2.0 ^b^	57.6 ± 1.6 ^b^
STR	82.5 ± 2.0 ^a^	73.0 ± 2.1 ^b^	71.8 ± 1.7 ^b^
WOB	87.0 ± 0.8 ^a^	82.1 ± 1.0 ^b^	80.1 ± 0.9 ^b^
ALH	2.7 ± 0.1	2.8 ± 0.1	3.1 ± 0.1
BCF	7.6 ± 0.2	7.7 ± 0.2	7.6 ± 0.1

Curvilinear velocity (VCL, µm/s), straight-line velocity (VSL, µm/s), average path velocity (VAP, µm/s), percentage of linearity (LIN, %), percentage of straightness (STR, %), wobble coefficient (WOB, %), mean amplitude of lateral head displacement (ALH, mm), and mean beat cross frequency (BCF, Hz). Data are depicted as means ± SEM. Different lowercase letters indicate differences among groups (*p* < 0.05).

**Table 4 animals-14-00914-t004:** Kinematic parameters of motility analyzed post-thaw between years and thawing methods (Year 0; Year 10, 37 °C; Year 10, 70 °C; *n* = 27) in Level 3.

	Year 0	Year 10, 37 °C	Year 10, 70 °C
VCL	99.1 ± 4.0 ^a^	83.9 ± 2.8 ^b^	90.3 ± 2.9 ^ab^
VSL	70.9 ± 2.9 ^a^	50.9 ± 1.3 ^b^	54.2 ± 1.4 ^b^
VAP	87.0 ± 3.4 ^a^	69.4 ± 2.4 ^b^	73.3 ± 2.4 ^b^
LIN	72.4 ± 1.8 ^a^	61.6 ± 1.7 ^b^	60.6 ± 1.4 ^b^
STR	82.2 ± 1.7 ^a^	74.6 ± 1.9 ^b^	74.7 ± 1.4 ^b^
WOB	88.0 ± 0.7 ^a^	82.6 ± 0.9 ^b^	81.2 ± 0.6 ^b^
ALH	2.7 ± 0.1 ^a^	2.8 ± 0.1 ^ab^	3.0 ± 0.1 ^b^
BCF	7.6 ± 0.1	8.0 ± 0.1	7.7 ± 0.1

Curvilinear velocity (VCL, µm/s), straight-line velocity (VSL, µm/s), average path velocity (VAP, µm/s), percentage of linearity (LIN, %), percentage of straightness (STR, %), wobble coefficient (WOB, %), mean amplitude of lateral head displacement (ALH, mm), and mean beat cross frequency (BCF, Hz). Data are depicted as means ± SEM. Different lowercase letters indicate differences among groups (*p* < 0.05).

## Data Availability

Data are contained within the article.

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
