# Peer review of "Evaluation of Different Thawing Protocols on Iberian Boar Sperm Preserved for 10 Years at Different Liquid Nitrogen Levels"

_animals, 2024, doi:10.3390/ani14060914_

Round 1
Reviewer 1 Report
Comments and Suggestions for Authors
The research, titled "Enhancing Sperm Quality in Long-Term Preservation of Iberian Boar Semen," aims to evaluate the effect of ten years of storage at different nitrogen levels and optimize thawing protocols for pig semen. The topic of the article is highly intriguing, particularly due to its potential implications for the conservation of animal genetic resources, especially for breeds with limited distribution or at risk of extinction. The paper aligns well with the aims of the journal. Here are some suggestions for the authors to enhance the quality of the article:
Rewrite the simple summary in accordance with the journal's instructions for authors (https://www.mdpi.com/journal/animals/instructions). It should include a clear statement of the problem addressed, the aims and objectives, pertinent results, conclusions from the study, and their societal value. Additionally, it should be written for a lay audience.
I suggest rewriting the Abstract section, including more numerical data and significances.
Consider excluding terms already present in the title from the keywords.
The materials and methods section needs improvement, including additional information on the animals used for semen collection in the study (e.g., average age, average weight, etc.). I also suggest mentioning whether the animals originated from a single breeding facility and the frequency of semen collection from the boars. Additionally, it might be interesting to include some data on the subjects' diet.
Enrich the discussion by addressing the study's limitations and practical applications.
In the discussion section, the authors can consider incorporating the use of lipidomics in evaluating semen quality, drawing insights from studies conducted in other species. Lipidomics, the comprehensive analysis of lipid species and their roles, has been increasingly recognized as a valuable tool in understanding sperm quality and functionality. By examining the lipid composition of sperm membranes, lipidomics can provide valuable insights into membrane stability, fluidity, and overall sperm health.
Here's a suggestion for integrating lipidomics into the discussion:
"In addition to the parameters assessed in our study, future research could explore the application of lipidomics in evaluating Iberian pig sperm quality. Lipidomics, which involves the comprehensive analysis of lipid species, has emerged as a promising approach in assessing semen quality in various species. Studies in other animal models have demonstrated the importance of lipid composition in sperm membranes for maintaining membrane integrity, fluidity, and ultimately sperm function. By analyzing the lipid profile of sperm samples, lipidomics can offer valuable insights into the physiological state of sperm cells and their susceptibility to cryopreservation-induced damage. Incorporating lipidomics into our evaluation may provide a more comprehensive understanding of the impact of long-term cryostorage and thawing protocols on Iberian pig sperm quality." (see: 10.3390/ANI13010008)
In the future perspective section, the authors could consider including elements related to the dissemination of their study's results, including the utilization of innovative tools such as social media, which have been demonstrated to be effective in scientific communication. Here's a suggestion for integrating this aspect into the future perspective:
"As we move forward, it is imperative to ensure that the findings of our study reach a wider audience within the scientific community and beyond. Effective dissemination of research results is key not only for advancing scientific knowledge but also for informing stakeholders and policymakers in the field of animal genetics and reproduction. In line with recent publications highlighting the importance of innovative communication strategies, we propose leveraging platforms such as social media to disseminate our findings to a broader audience.
Social media platforms offer a unique opportunity to engage with researchers, practitioners, and the general public in real-time. By sharing key insights, updates, and implications of our study on platforms like Twitter, LinkedIn, and ResearchGate, we can foster discussions, facilitate collaboration, and raise awareness about the importance of genetic resource conservation in pig breeds, particularly the Iberian pig. Additionally, utilizing multimedia formats such as infographics, videos, and podcasts can enhance the accessibility and impact of our research findings, making them more understandable and relatable to diverse audiences. See: 10.3390/ani13223503
Furthermore, collaboration with science communication experts and science communicators can help tailor our messages effectively and ensure that they resonate with different stakeholders. By embracing innovative communication tools and strategies, we can maximize the reach and relevance of our research, ultimately contributing to informed decision-making and sustainable practices in animal breeding and conservation."
I kindly suggest expanding the conclusions section of your paper to provide a more detailed and comprehensive report of the main findings. This will help readers better understand the significance of your research.
Please double-check the reference list to ensure that all references are included in the main text and vice versa.
Author Response
Referee: 1
The research, titled "Enhancing Sperm Quality in Long-Term Preservation of Iberian Boar Semen," aims to evaluate the effect of ten years of storage at different nitrogen levels and optimize thawing protocols for pig semen. The topic of the article is highly intriguing, particularly due to its potential implications for the conservation of animal genetic resources, especially for breeds with limited distribution or at risk of extinction. The paper aligns well with the aims of the journal. Here are some suggestions for the authors to enhance the quality of the article:
Reply: We appreciate the constructive feedback provided by the referee and we have addressed all the potential concerns raised by the reviewer.
Rewrite the simple summary in accordance with the journal's instructions for authors (https://www.mdpi.com/journal/animals/instructions). It should include a clear statement of the problem addressed, the aims and objectives, pertinent results, conclusions from the study, and their societal value. Additionally, it should be written for a lay audience.
Reply: We thank the referee for their suggestion. We have changed the summary abstract to provide clearer elucidation of the problem, objectives, and results of the study.
I suggest rewriting the Abstract section, including more numerical data and significances.
Reply: We appreciate the reviewer's suggestion, and we recognize that including numerical data in the abstract could enhance the comprehensiveness of the information provided. However, due to the space limitations of the abstract and the extensive amount of information required to adequately describe the study design and results, incorporating numerical data may result in the loss of crucial details necessary for a comprehensive understanding of the work. Therefore, we have opted to prioritize clarity and succinctness in conveying the essence of the study within the confines of the abstract.
Consider excluding terms already present in the title from the keywords.
Reply: The keywords have been modified following the suggested change in the title by the other reviewers.
The materials and methods section needs improvement, including additional information on the animals used for semen collection in the study (e.g., average age, average weight, etc.). I also suggest mentioning whether the animals originated from a single breeding facility and the frequency of semen collection from the boars. Additionally, it might be interesting to include some data on the subjects' diet.
Reply: We have supplemented this information in the Materials and Methods section in response to your queries and those of the other reviewers. Specifically addressing your questions:
The average age and origin were previously mentioned in lines 109 and 110 (age-range of 1.5 and 2 years and were housed at different Spanish AI centers.) but we have rephrased it for better clarity.
The data regarding weights and feed types were not provided by the AI centers from which the ejaculates were obtained, but all animals were maintained on an isoenergetic and isoproteic diet appropriate for this breed.
Enrich the discussion by addressing the study's limitations and practical applications.
Reply: This has been included in the conclusion.
In the discussion section, the authors can consider incorporating the use of lipidomics in evaluating semen quality, drawing insights from studies conducted in other species. Lipidomics, the comprehensive analysis of lipid species and their roles, has been increasingly recognized as a valuable tool in understanding sperm quality and functionality. By examining the lipid composition of sperm membranes, lipidomics can provide valuable insights into membrane stability, fluidity, and overall sperm health.
Here's a suggestion for integrating lipidomics into the discussion:
"In addition to the parameters assessed in our study, future research could explore the application of lipidomics in evaluating Iberian pig sperm quality. Lipidomics, which involves the comprehensive analysis of lipid species, has emerged as a promising approach in assessing semen quality in various species. Studies in other animal models have demonstrated the importance of lipid composition in sperm membranes for maintaining membrane integrity, fluidity, and ultimately sperm function. By analyzing the lipid profile of sperm samples, lipidomics can offer valuable insights into the physiological state of sperm cells and their susceptibility to cryopreservation-induced damage. Incorporating lipidomics into our evaluation may provide a more comprehensive understanding of the impact of long-term cryostorage and thawing protocols on Iberian pig sperm quality." (see: 10.3390/ANI13010008)
Reply: Once again, we express our gratitude for the constructive comments provided by the referee. In response to this feedback, we have incorporated a paragraph into the discussion section. The referee is correct in noting that the analysis of membrane lipids can offer an alternative perspective on the results of cryopreservation, as the corresponding author of this article has seen study, whose 2011 doctoral thesis had a primary focus on evaluating the lipid composition of the sperm membrane in Iberian pigs and its association with semen freezability.
In the future perspective section, the authors could consider including elements related to the dissemination of their study's results, including the utilization of innovative tools such as social media, which have been demonstrated to be effective in scientific communication. Here's a suggestion for integrating this aspect into the future perspective:
"As we move forward, it is imperative to ensure that the findings of our study reach a wider audience within the scientific community and beyond. Effective dissemination of research results is key not only for advancing scientific knowledge but also for informing stakeholders and policymakers in the field of animal genetics and reproduction. In line with recent publications highlighting the importance of innovative communication strategies, we propose leveraging platforms such as social media to disseminate our findings to a broader audience.
Social media platforms offer a unique opportunity to engage with researchers, practitioners, and the general public in real-time. By sharing key insights, updates, and implications of our study on platforms like Twitter, LinkedIn, and ResearchGate, we can foster discussions, facilitate collaboration, and raise awareness about the importance of genetic resource conservation in pig breeds, particularly the Iberian pig. Additionally, utilizing multimedia formats such as infographics, videos, and podcasts can enhance the accessibility and impact of our research findings, making them more understandable and relatable to diverse audiences. See: 10.3390/ani13223503
Furthermore, collaboration with science communication experts and science communicators can help tailor our messages effectively and ensure that they resonate with different stakeholders. By embracing innovative communication tools and strategies, we can maximize the reach and relevance of our research, ultimately contributing to informed decision-making and sustainable practices in animal breeding and conservation."
Reply: We fully acknowledge the reviewer's perspective and the insights presented in the referenced paper (DOI: 10.3390/ani13223503). However, it's important to note that the specific focus of the work highlighted by the reviewer is on utilizing these tools for disseminating scientific findings. While our study does not have that precise objective, if published, it will be disseminated extensively. The researchers involved in our study maintain active accounts on these social networks, and the institution affiliated with the CSIC (Spanish National Research Council) has a robust policy for advertising and disseminating research outcomes. More detailed information regarding this policy can be found at the following link: ttps://www.csic.es/es/investigacion/catalogo- of-scientific-technical-services/service-units/dissemination. Additionally, the CSIC enjoys a large following on its institutional accounts, ensuring that research findings are accessible to both the general public and fellow researchers and collaborators, as noted by the reviewer.
I kindly suggest expanding the conclusions section of your paper to provide a more detailed and comprehensive report of the main findings. This will help readers better understand the significance of your research.
Reply: We have enhanced the conclusion section to provide more comprehensive information.
Please double-check the reference list to ensure that all references are included in the main text and vice versa.
Reply: We have reviewed the references to ensure their accuracy and completeness.
Reviewer 2 Report
Comments and Suggestions for Authors
In the reviewed study, the authors showed that LN2 levels fluctuation during 10 years of cryostorage did not influence Iberian pig sperm quality. Thawing at 70°C, improve semen quality especially in samples constantly submerged in liquid nitrogen, compared to those thawed traditionally at 37°C. This observation is important, but such modification of thawing procedure has already been described by the same authors. In general, the study doesn’t’ contribute much, however there are few studies analyzing semen quality after such a long time of storage.
Overall, the manuscript is well written, however the introduction laks information on the achievements and problems with cryopreservation of boar semen..
My biggest doubts are the analyzes regarding Year 0.
Detailed comments:
Lines 59-60 – when writing about boar semen, the authors refer to literature [5, 8,10] - of which [8] and [10] concern bull semen, and only one [5] - boar semen - please provide appropriate literature
Line 64: [12: Dalton] and References: - please check the bibliographic data - Dalton is not the first author of this publication
M&M
Line 106: we can read that: For the study 53 ejaculates from 53 different boars were used - does this mean that each frozen straw/sample came from the ejaculate of a different male?
- how many straws were frozen from one ejaculate, why only one was used and why the numbers of straws at each nitrogen level were not the same? - please clarify
Lines 128-129: Year 0 - please clarify how many straws/samples in total were assessed one week after freezing (n=3 ? i.e. one from each NL2 level?)
- if at the beginning of the experiment the entire storage tank was filled with liquid nitrogen, it can be assumed that after a week of storage, the straws/samples at all levels were still completely immersed in liquid nitrogen. Therefore, in my opinion for Year 0, both in Table 1 and Figure 2 - the results should be presented as an average value of three samples/straws for each semen parameter analysed .
lines 145-150 – if one straw was removed from each level after one week of storage, the number of samples assessed after 10 years was lower by 1 than – please specify the number of samples assessed at each level after 10 years of storage.
Line 185-190: Acrosome status assessment - it is not clear how the integrity of the acrosome was assessed, even in the paper referred to by the authors [15], no detailed information about the methodology was provided. Currently specific markers such e.g. FITC-PSA, FITC-PSA/PI or Alexa Fluor 488 Conjugate and flow cytometer analysis are usually used for acrosome status assessment
Results:
Fig 2: Year 0 – if one sample was analyzed from each LN2 level, please explain how the error bar for such a sample was calculated ?
- the number of samples thawed at 37°C and 70°C should be provided for each level
Fig. 3, Fig. 4, Fig. 5 (similar to Fig 2) – if one sample was analyzed at Year 0 from each LN2 level, please explain how the error bar for such a sample was calculated ?
-in each Figure, the number of samples thawed at 37°C and 70°C should be provided
Author Response
Referee: 2
Comments to the Author
In the reviewed study, the authors showed that LN2 levels fluctuation during 10 years of cryostorage did not influence Iberian pig sperm quality. Thawing at 70°C, improve semen quality especially in samples constantly submerged in liquid nitrogen, compared to those thawed traditionally at 37°C. This observation is important, but such modification of thawing procedure has already been described by the same authors. In general, the study doesn’t’ contribute much, however there are few studies analyzing semen quality after such a long time of storage.
Overall, the manuscript is well written, however the introduction laks information on the achievements and problems with cryopreservation of boar semen..
Reply: Thank you very much for your suggestion. You are correct that additional information about freezing systems, their improvements, and problems could enhance the introduction. However, since the primary objective of this study was to evaluate thawing and conservation status, and to prevent the introduction from becoming overly lengthy, we chose to focus solely on these two aspects.
My biggest doubts are the analyses regarding Year 0. Detailed comments:
Lines 59-60 – when writing about boar semen, the authors refer to literature [5, 8,10] - of which [8] and [10] concern bull semen, and only one [5] - boar semen - please provide appropriate literature
Reply: The reviewer is right. This has been corrected with proper references
Line 64: [12: Dalton] and References: - please check the bibliographic data - Dalton is not the first author of this publication
Reply: We are sorry for the mistake; we have corrected this reference.
M&M
Line 106: we can read that: For the study 53 ejaculates from 53 different boars were used - does this mean that each frozen straw/sample came from the ejaculate of a different male?
Reply: Yes, it is a single ejaculate from each of the 53 animals. We have changed this in the text for better understanding
- how many straws were frozen from one ejaculate, why only one was used and why the numbers of straws at each nitrogen level were not the same? - please clarify
Reply: We apologize for any confusion caused by our previous text. The quantity of frozen straws per ejaculate varied due to differences in volume and concentration across ejaculates. It's important to note that this bank serves as an official repository for the conservation of animal genetic resources. Upon freezing, the animals are individually identified, and their straws are arranged on racks, with each animal being at a unique level, from bottom to top
Lines 128-129: Year 0 - please clarify how many straws/samples in total were assessed one week after freezing (n=3 ? i.e. one from each NL2 level?)
Reply: During both Year 0 and Year 10, two straws from each animal at every level were thawed and analyzed, totalling 9 animals at Level 1, 17 at Level 2, and 27 at Level 3. This has been indicated in the text and in the figures and tables.
- if at the beginning of the experiment the entire storage tank was filled with liquid nitrogen, it can be assumed that after a week of storage, the straws/samples at all levels were still completely immersed in liquid nitrogen. Therefore, in my opinion for Year 0, both in Table 1 and Figure 2 - the results should be presented as an average value of three samples/straws for each semen parameter analysed .
Reply: In this regard, we respectfully disagree with the reviewer's suggestion. Our approach was not to perform direct comparisons between individual animals across different levels. Rather, our experimental design focuses on comparing outcomes across levels over time. Each level serves as our experimental unit. Before our analyses, we ensured that there were no significant differences in the measured variables among the different levels at Year 0. Thus, although comprised of different animals, all levels commenced from a comparable baseline of seminal quality.
lines 145-150 – if one straw was removed from each level after one week of storage, the number of samples assessed after 10 years was lower by 1 than – please specify the number of samples assessed at each level after 10 years of storage.
Reply: At both Year 0 and Year 10, during each thawing process, one boar (2 straws) from each level was thawed.
Line 185-190: Acrosome status assessment - it is not clear how the integrity of the acrosome was assessed, even in the paper referred to by the authors [15], no detailed information about the methodology was provided. Currently specific markers such e.g. FITC-PSA, FITC-PSA/PI or Alexa Fluor 488 Conjugate and flow cytometer analysis are usually used for acrosome status assessment
Reply: The reviewer is correct. Presently, there exist more advanced methods for assessing morphology compared to those employed in this study. However, it's important to note that 10 years ago, in our experimental design the acrosome status of the samples was evaluated using phase contrast acrosome morphological analysis. To maintain consistency in the analysis over the 10-year preservation period and to avoid introducing bias due to changes in techniques, this method was retained. Although this technique is becoming less common in scientific research, it remains a validated and routinely utilized method, particularly in field conditions where access to fluorescence microscopy may be limited. Detailed descriptions and phase contrast photographs of the various morphologies analyzed in this study can be found in the referenced article.
Results:
Fig 2: Year 0 – if one sample was analyzed from each LN2 level, please explain how the error bar for such a sample was calculated ?
Reply: Each level has a number of animals, and all animals of each level were analyzed.
- the number of samples thawed at 37°C and 70°C should be provided for each level
Reply: This has been modified accordingly in the current version of the text
Fig. 3, Fig. 4, Fig. 5 (similar to Fig 2) – if one sample was analyzed at Year 0 from each LN2 level, please explain how the error bar for such a sample was calculated ?
Reply: Once again, we apologize if this was not adequately explained in the text. As previously described, each level has a number of animals, and all animals of each level were analyzed. We have elaborated further on the information in the text to provide greater clarification.
-in each Figure, the number of samples thawed at 37°C and 70°C should be provided
Reply: The n of each level and year has been included in the figure legend and tables.
Reviewer 3 Report
Comments and Suggestions for Authors
please see the attached file.

minor.
Author Response
Referee: 3
Comments to the Author
The present study evaluated the influence of cryostorage time and semen straw position in the LN tank on Iberian boar sperm quality, and modified the thawing methods with an aim to improve the quality of post-thaw sperm. The results showed no significant impact of storage time and straw position in LN on sperm quality, but the modified thawing procedure improved the frozen-thawed sperm quality. This study provides useful information for routine sperm bank management and technical support in boar sperm thawing procedure for achieving better results. There are several points regarding the contents of this study I would like to share my opinions with the authors.
Reply: Thanks for your time and critical comment to our manuscript. We included in here a detailed response to your comments as well as changes highlighted in the new version of the manuscript.
General comments:
This study showed a good presentation of the results, which will provide useful information for sperm bank management and promote the use of frozen boar semen in practice by improving thawing method. However, the title is suggested to cover all the contents presented, including the evaluation of cryostorage time and liquid nitrogen levels on post-thaw boar sperm quality, and the results from the modification of thawing procedure.
Reply: Thank you for your input. In consideration of your feedback, along with that of another reviewer, the title has been adjusted to improve the comprehensibility of the study.
In addition, as no significant impact of cryostorage on boar sperm quality was found, why the authors made further efforts to improve the post-thaw sperm quality?
Reply: Our research group has demonstrated in various scientific articles how modifying the thawing protocol can enhance the quality of frozen samples. We deemed it important to evaluate this phenomenon in samples preserved for 10 years, given the controversy that has arisen in recent years regarding the potential degradation of cryopreserved samples over time (an issue addressed in the introduction and discussion). Our aim was twofold: 1) to ascertain whether the experimental conditions we proposed, which involved samples either fully immersed or not fully immersed in liquid nitrogen, could be a determining factor, and 2) if a problem were to exist, to ascertain whether modifying the protocol could help prevent or mitigate such damage occurred over time.
For a retrospective study, it is better to have the samples as many as possible. It seems very few with 53 ejaculates used in this study.
Reply: A total of 53 ejaculates, each from a distinct animal, were utilized in the study. Considering the limited availability of male specimens within the Iberian pig breed, this sample size is notably substantial. Moreover, it is noteworthy that the number of animals utilized in our study is comparable to that employed by Li et al., 2018, and significantly exceeds the sample sizes reported in other studies such as Fraser et al., 2014 (with 10 animals), or Kozumplík, 1985 (with 4 animals), which are the only studies identified where the extended preservation of samples in liquid nitrogen has been investigated in porcine species.
In the results, using 70 degrees for 8 s achieved better sperm quality from samples stored at level 3, why is that? If the modified thawing procedure works well, it should work well to all the samples, not just those from level 3 constantly immerged in LN, right?
Reply: The reviewer's observation is insightful. As commented in our discussion, our experimental design suggests that the observed improvement predominantly occurred in the lower levels. This phenomenon could potentially be attributed to fluctuations in temperature within the tank, corresponding to variations in nitrogen levels. It is plausible that samples situated at higher levels experienced gradual temperature increases as the nitrogen level depleted, which would reverse upon tank refilling. This dynamic may have facilitated processes of de-crystallization and re-crystallization, particularly pronounced in level 1, less so in level 2, and negligible in level 3. Notably, while these processes do not induce lethal damage to sperm, as evidenced by the absence of significant differences in acrosome state, viability, or fragmentation after ten years in level 1, they do cause irreversible damage that cannot be rectified or prevented during the thawing process.
Specific comments:
Q1. Line 10, as cryopreservation of gametes is just one of the methods used to conserve genes, the expression “Cryopreservation of gametes is the basis of the establishment of a genetic resource bank” can be modified.
Reply: Thank you for your feedback. At the request of another reviewer, the abstract has been rewritten, and we have taken your comment into consideration.
Q2. Line 17, “nitrogen levels variations” should be specified, for example, changed to “variations in liquid nitrogen levels where frozen boar semen was stored”. All the “nitrogen” mentioned in the text should be replaced by “liquid nitrogen”.
Reply: Once again, thank you for your input, which has been incorporated into both the abstract and we have included the abbreviation LN2 for liquid nitrogen in most of the text.
Q3. Line 19, when we talk about the thawing methods, both thawing temperature and time should be included. Please add the information of the modified and the routine thawing process. So is the case in line 29, 32 and 36.
Reply: This has been included and modified accordingly.
Q4. Line 10 and line 20, there are errors in the sentences. There should be an of behind establishment in line 10 and “stores” should be replaced with “stored” in line 20. Similar cases can be found elsewhere in the following text. Please check it carefully.
Reply: Thank you once more for your feedback, which has been considered in the revised abstract and reviewed throughout the text.
Q5. Line 28, only one straw from each ejaculates was thawed and analyzed, which could bring fluctuations in results. Line 128, how many straws were made from each ejaculate? how many straws from each ejaculate were thawed for the experiment? Did the authors make pooled semen?
Reply: We apologize for the lack of clarity in the text. Contrary to what was stated, only one sample per male was thawed, with each male having two straws per sample to mitigate the potential impact of any straw-specific effects.
The quantity of frozen straws per ejaculate varied due to differences in volume and concentration across ejaculates. Given the genetic significance of the animals involved, we aimed to minimize the number of straws thawed for testing purposes. While there may be inherent differences in quality among straws, it's worth noting that thawing multiple straws under high defrosting temperatures and short durations can introduce additional variability. In previous study, we observed that even slight variations in thawing duration (e.g., 1-2 seconds) could impact quality. Consequently, to conserve as many doses as possible from these valuable animals and mitigate potential variability, we opted to thaw only two straw from each animal in each experiment.
Q6. Line 160, the version of the CASA used should be provided.
Reply: We have included the CASA version in the Materials and Methods section.
Q7. In all the tables, all the values presented should keep the same decimal number. For all the figures, it is suggested to add the legend just beside the figure to reach an easy reading. It is recommended to present the results of all the three LN levels using line graph, which could clearly depict the changes of sperm quality with time.
Reply: The tables have been reviewed, and figures have been added with accompanying legends to enhance comprehension. However, we believe that employing a line graph may potentially lead to misinterpretation, since there are only two distinct time points (year 0 and year 10), but with two thawing methods in year 10.
Q8. Line 427-431, actually the modified thawing procedure requires skilled operation. The fine control of thawing temperature and time is the key to achieve the best results. It is not convenient to handle for normal staff working in the pig farm. So, it is suggested to highlight the best results achieved by the modified protocol, without stating the convenience in the practice.
Reply: The reviewer is correct; working with such high temperatures and short times can be challenging. As a result, the conclusions have been modified.
The English has been reviewed and modified by an English expert to identify and correct any potential writing errors.
Reviewer 4 Report
Comments and Suggestions for Authors
The article focuses on investigating differences in sperm motility parameters in sperm from Iberian boar -stored at 0 and 10 years in Liquid Nitrogen. It also researched the impact of depth/location of the straws on parameters such as Plasma membrane integrity, Acrosome integrity and DNA fragmentation. The study found differences in the different levels of storage in liquid nitrogen.
1. The title of the paper is broad and should be narrowed to describe the variables considered in the article. Therefore, the topic should be modified.
2. There are some missing information in the methodology. For example, were the ejaculates pooled before they were placed in straws? What Diluent was used in diluting the samples at year 0? If the diluent were different such differences could account for differences in the results. What is the control sample? How many times was the samples replicated?
Comments on the Quality of English LanguageKindly have a native speaker edit the article. There are some run on sentences and tense issues.
Author Response
Referee: 4
Comments to the Author
The article focuses on investigating differences in sperm motility parameters in sperm from Iberian boar -stored at 0 and 10 years in Liquid Nitrogen. It also researched the impact of depth/location of the straws on parameters such as Plasma membrane integrity, Acrosome integrity and DNA fragmentation. The study found differences in the different levels of storage in liquid nitrogen.
Reply: We thank the through revision that he/she did of our manuscript. We included every suggestion he/she pointed out throughout the manuscript
- The title of the paper is broad and should be narrowed to describe the variables considered in the article. Therefore, the topic should be modified.
Reply: We have revised the title based on the feedback provided by reviewers 2 and 3, aiming for improved completeness and conciseness.
- There are some missing information in the methodology. For example, were the ejaculates pooled before they were placed in straws? What Diluent was used in diluting the samples at year 0? If the diluent were different such differences could account for differences in the results. What is the control sample? How many times was the samples replicated?
Reply: We have completed this information in the material and methods according to your questions and those of the other reviewers.
Were the ejaculates collected before placing them in straws?
Reply: No, the frozen ejaculates were from individual animals.
What diluent was used to dilute the samples in year 0? If the diluent were different, such differences could explain differences in the results.
Reply: This is a good question, we have completed this information. But specifically, prior to freezing, all the ejaculates once obtained were diluted 1:1 and stored at 17ºC in BTS extender until frozen. The BTS used was the same extender that was subsequently used for thawing.
What is the control sample?
Reply: No explicit control group is employed in this study, as its primary objective is to assess frozen semen samples both at the onset of storage (year 0) and after a decade. The core inquiry pertains to the potential degradation of semen quality over the ten-year period, as suggested by existing literature. Additionally, the investigation incorporates a validated protocol modification aimed at enhancing the thawing process. This intervention seeks to ascertain whether any perceived decline in semen quality for the long stored is mitigatable or preventable through optimization of thawing techniques.
How many times were the samples replicated?
Reply: The ejaculates utilized in the study comprised 53 samples obtained from 53 distinct animals, each subjected to a single freezing event. We consider that this is a representative number in this native pig breed, since the number of males of this breed is very small compared to other white pigs’ breeds.
Comments on the Quality of English Language:
Kindly have a native speaker edit the article. There are some run on sentences and tense issues.
Reply: Thanks for your suggestion. The English has been reviewed and modified by an English expert to identify and correct any potential writing errors.
Round 2
Reviewer 1 Report
Comments and Suggestions for Authors
the paper improved a lot
Reviewer 4 Report
Comments and Suggestions for Authors
Thanks for addressing the concerns raised.
Comments on the Quality of English LanguageThe English Language quality has been improved